# Goal congruency dominates reward value in accounting for behavioral and neural correlates of value-based decision-making

Romy Frömer [1]*, Carolyn K. Dean Wolf[1] & Amitai Shenhav[1]*

When choosing between options, whether menu items or career paths, we can evaluate how rewarding each one will be, or how congruent it is with our current choice goal (e.g., to point out the best option or the worst one.). Past decision-making research interpreted findings through the former lens, but in these experiments the most rewarding option was always most congruent with the task goal (choosing the best option). It is therefore unclear to what extent expected reward vs. goal congruency can account for choice value findings. To deconfound these two variables, we performed three behavioral studies and an fMRI study in which the task goal varied between identifying the best vs. the worst option. Contrary to prevailing accounts, we find that goal congruency dominates choice behavior and neural activity. We separately identify dissociable signals of expected reward. Our findings call for a reinterpretation of previous research on value-based choice.

[1] Cognitive, Linguistic, and Psychological Sciences, Carney Institute for Brain Science, Brown University, Providence, RI, USA. *email: romy_fromer@brown.edu; amitai_shenhav@brown.edu

Reward is central to the behavior of humans and animals alike[1]. We repeat rewarded behavior and strive to select actions that maximize reward[2,3]. Reward therefore serves dual roles, as an end unto itself and as a guide towards a particular goal. The significance of this distinction has often been overlooked.

A large body of research has examined how we evaluate the expected reward for an object or a course of action, including how this reward value is shaped by feature integration[4,5], context[6,7], and motivational state[8]. When people choose between multiple options with varying reward, these studies have demonstrated consistent behavioral and neural correlates of those reward values. People make faster decisions when the option they choose is much more rewarding than the option(s) they forego[9,10], and when the overall (i.e., average) reward associated with those choice options is high[11–14]. These behavioral findings are paralleled by increasing relative and overall value related activity in regions of a well-characterized value network[15,16], that includes regions of ventral striatum and ventromedial prefrontal cortex. These behavioral and neural indices have thus been taken to reflect variability in how rewarding one's options are[3]. However, this interpretation has overlooked a confound common to almost all such experiments, between an option's promise of reward and its ability to support a person's task goal.

Specifically, participants in studies of value-based decision-making are typically instructed to choose the item they prefer most[9,17–21] or dislike least[21–25]. As a result, the option that is most rewarding or least aversive also represents the option that is most congruent with the participant's choice goal. The relative reward value of a choice set (how much more rewarding the chosen option is than the alternatives) is therefore identical to the relative goal value of those options (how much more goal-congruent the chosen option is than the alternatives). Having more rewarding options in one's choice set (higher overall reward value) also necessarily means having more options that are congruent with one's goal (higher overall goal value). It is thus unclear whether behavioral findings previously attributed to reward value, such as the choice speeding effect of having options that are overall more rewarding, can instead be accounted for by the goal congruency of these reward values. It is similarly unknown to what extent putative neural correlates of relative and overall reward value instead reflect relative and overall goal congruency. The present studies aim to address these open questions.

There are several a priori reasons to favor a strictly reward-based interpretation of past research on choice value. First, research on Pavlovian conditioning shows that animals have an innate drive to approach rewards and avoid punishments, such that they are faster to approach a reward than to avoid it[26,27]. Second in an operant task (e.g., lever-pressing), the vigor (and therefore speed) with which an animal responds scales with the average rate of reward in their environment[28–30]. Third, there is ample evidence that a network of brain regions – the value network – processes both anticipated and experienced reward[15,31]. For instance, it has been shown that activity in this network correlates with the reward value assigned to individual items[32–35] and sets of items[10,16,21,36], and that reward-related activity associated with those items can be used to predict one's choice between multiple such options[31,37,38]. As a result, value network activity is generally assumed to encode a form of reward value, one that is multifaceted (varying, e.g., with one's motivational state[8] and context[7,39–41]) but should remain directionally consistent across choice goals. That is, an item with high reward value should be encoded similarly by this network whether an individual is asked to indicate that they like that item or that they do not dislike that item[42,43]. However, previous research on choice-related behavior and neural activity have failed to dissociate reward-centered accounts of these findings from accounts that are centered on the congruency of those rewards with one's current goal.

Research on recognition memory provides an excellent example of how goal congruency confounds can engender misinterpretations of a set of findings. Studies of frequency discrimination required participants to indicate which of a pair of items had been presented more frequently during an earlier study phase[44,45]. Participants responded faster when both options were high frequency items than when both were low frequency items, suggesting that response speed was correlated with the overall frequency of the choice set. However, this effect reversed when participants were instructed to choose the less frequent item; under these instructions, participants were instead faster to respond when the overall frequency was low. Thus, by manipulating choice instructions, this study dissociated item frequency from goal congruency, and showed that behavior was driven by the latter rather than the former.

Here, we employ a similar approach, asking participants to either choose the best (most rewarding) or the worst (least rewarding) item in a value-based choice set, to test whether task goals modulate the relationship reward value has with behavior and brain activity. Across four studies, we find that a participant's decision speed is not accounted for by the reward value of their options but rather the goal congruency of those options, including how congruent those options are overall (a feature of the choice set that is irrelevant to goal achievement). We demonstrate that these findings can be captured by modifying an established computational model of choice dynamics to account for an individual's current choice goal. We further find that the brain's value network tracks the relative value of a chosen item only in terms of its goal congruency, while at the same time tracking both the overall goal congruency and overall reward associated with the choice set. Within striatum, correlates of (overall) reward value and (relative) goal congruency were spatially dissociated. Our findings suggest that, in common choice paradigms, reward's relationship to behavior and neural activity is less direct than previously thought, calling for a reinterpretation of previous findings in research on value-based decision-making.

## Results

**Task overview.** Participants in two studies (Ns = 30) made a series of hypothetical choices between sets of four consumer goods (Fig. 1a). Choice sets were tailored to each participant, based on how rewarding they had rated each item individually earlier in the session, and these sets varied in their overall reward value (how rewarding the items are on average) and in the similarity of option values to one another (Supplementary Fig. 4). Participants in Study 2 performed this choice task while undergoing fMRI.

**Goal congruency drives overall value effects on choice speed.** Previous studies have consistently shown that participants are faster to select their most preferred item out of a set as the overall reward value of their options increases[11–14]. These findings can be explained by accounts that posit a direct relationship between potential rewards and choice speed (e.g., Pavlovian accounts) or they could be explained by the fact that reward value was always congruent with one's task goal in these studies (to choose the most rewarding item). To test these competing accounts, we explicitly varied this goal while participants made choices between sets of options. For half of the trials, we instructed participants to choose their most preferred from each choice set (i.e., the item with the highest reward value; Choose Best task). In

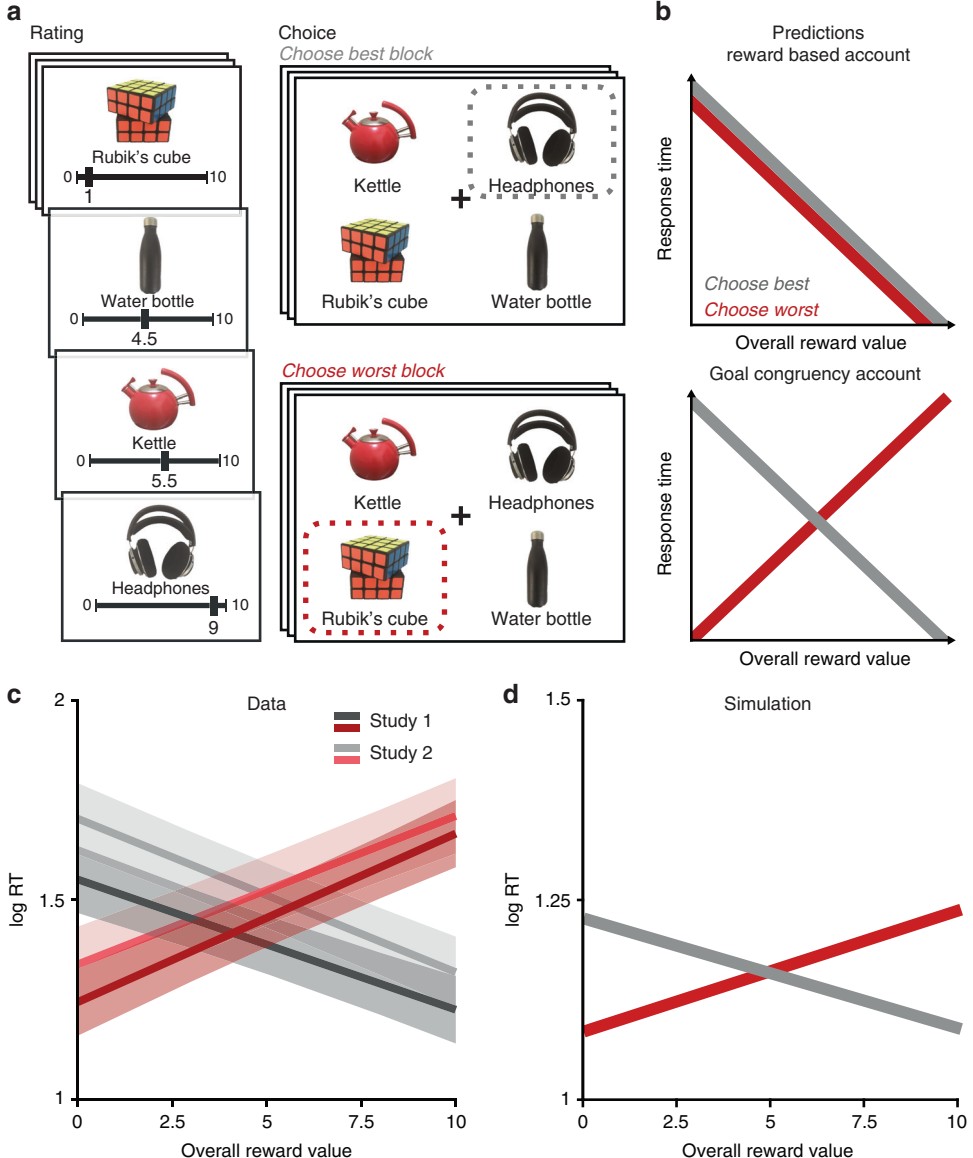

**Fig. 1** Overall value effects on RT are driven by goal congruency rather than reward value. **a** After evaluating each item in isolation (left), participants saw sets of four options and (in separate blocks) were instructed to choose either the best or the worst option (right). The same example is shown for both blocks but each choice sets was only viewed once in a session. **b** Top: A reward-based account predicts that RTs should decrease with overall value of the set, irrespective of the choice goal. Bottom: A goal congruency account predicts that RTs should decrease with overall value in Choose-Best blocks but instead increase with overall value in Choose-Worst Blocks. **c** Both Study 1 (behavioral) and Study 2 (fMRI) find the task-specific RT reversal predicted by a goal congruency account (see also Supplementary Study 1, Supplementary Discussion). Shaded error bars show 95% confidence intervals. **d** Our empirical findings were captured by an LCA model that took goal values (rather than reward values) as inputs

this task (the standard one used in research on value-based choice), there is a positive relationship between reward value and goal congruency (which we will also refer to as goal value) – the more rewarding an item, the more congruent it is with the task goal. For the other half of trials, we reversed this instruction, asking participants to instead choose their least preferred from a given set (i.e., the item with the lowest reward value; Choose Worst task). Unlike the Choose Best task, for the Choose Worst task there was a negative relationship between reward value and goal congruency (i.e., the least rewarding items were now the most congruent with the task goal).

If choices are sped up by the overall reward value of a choice set (as inferred by previous research), this should be true irrespective of whether participants were performing the Choose Best or Choose Worst task (Fig. 1b, top). If these previous findings instead reflect variability in choice speed with goal congruency, then RTs should decrease with overall reward value in the Choose Best condition but increase with overall reward value in the Choose Worst condition (Fig. 1b, bottom).

Across both studies, we found strong evidence for a goal congruency account of choice behavior. As an initial test, we ran linear mixed effects models separately on Choose-Best and Choose-Worst trials. Consistent with previous studies which instructed participants to choose the best item in a set[11–14], we found that RTs decreased with overall reward value for our Choose-Best task (Study 1: $b = -0.33$, $p < 0.001$, Study 2: $b = -0.39$, $p < 0.001$). However, for the Choose-Worst task, we found that this relationship flipped – RTs instead increased rather than decreased with overall value (Study 1: $b = 0.42$, $p < 0.001$, Study 2: $b = 0.38$, $p < 0.001$). Across all trials, RTs therefore demonstrated

an interaction between choice goal and overall reward value, as predicted by a goal congruency account (Fig. 1c; Table 1, top; Study 1: $b = -0.75$, $t = -12.66$, $p < 0.001$, Study 2: $b = -0.76$, $t = -14.39$, $p < 0.001$). Once accounting for this interaction, we did not find residual main effects of overall value across the two tasks (Study 1: $t = 1.61$, $p = 0.108$; Study 2: $t = -0.24$, $p = 0.809$), nor did we find a main effect of choice goal (Study 1: $t = -1.94$, $p = 0.062$; Study 2: $t = -0.01$, $p = 0.736$). In other words, RTs varied with the overall reward value inversely and symmetrically between the two tasks, but choices were not overall faster for one task or another.

This pattern of findings is qualitatively consistent with the prediction that overall reward value influences choice RT as a function of its goal congruency rather than its association with potential rewards. In order to test this interpretation directly, we constructed an additional overall value variable that encoded the overall goal congruency of one's options (overall goal value). Overall goal value increases with overall reward value in the Choose-Best condition and decreases with overall reward value in the Choose-Worst condition. We compared separate linear mixed effects models predicting choice RT by overall reward value alone (Model 1), overall reward value in interaction with task condition (Model 2), or overall goal value alone (Model 3). We found that Models 2 and 3 substantially outperformed Model 1 (Table 2). Accordingly, when including overall goal value and overall reward value in the same regression, we find a significant effect only for overall goal value (Study 1: $b = -0.37$, $p < .001$; Study 2: $b = -0.38$, $p < 0.001$) and not for overall reward value (Study 1: $b = 0.05$, $p = 0.108$; Study 2: $b = -0.01$, $p = 0.809$; Table 1, bottom).

This effect of overall value on choice RT is independent of the effect of value difference, i.e., the absolute difference between the goal value (highest item value for Choose-Best and lowest item value for Choose-Worst) and the average value of the remaining items. As in previous studies, we find that easier choices (those with higher value differences) are faster (Table 1), an effect that we control for in all of our analyses.

Across both studies, analyzing choices as a function of value difference, overall value, task and their interactions, we find that value difference also predicts the consistency (cf. accuracy) of choices – as value difference increased, participants were more likely to choose the item that achieved their current choice goal (the highest-value item in Choose-Best, the lowest-value item in Choose-Worst; Study 1: $b = 3.76$, $z = 13.12$, $p < 0.001$, Study 2: $b = 3.65$, $z = 13.40$, $p < 0.001$). The influence of value difference on choice accuracy did not differ between the two choice goals (Study 1: $b = 0.58$, $z = 1.01$, $p = 0.311$, Study 2: $b = 0.53$, $z = 0.98$, $p = 0.326$). Unlike value difference, the overall value of a set did not significantly predict choice consistency for either task (Study 1: $b = 0.33$, $z = 1.08$, $p = .280$, Study 2: $b = 0.49$, $z = 1.70$, $p = 0.090$).

The behavioral findings we observe across Studies 1 and 2 raise two potential concerns that we addressed in follow-up studies. First, it could be the case that the novel findings we observed in the Choose-Worst condition benefited from participants having the (traditional) Choose-Best condition as a within-session reference point. While the counter-balancing of these tasks mitigates this possibility, to rule it out we performed a follow-up study in which participants only performed the Choose-Worst task (Supplementary Study 1), and replicated the overall value findings we observed for this condition in Study 1 (Supplementary Discussion 1). Second, it is possible that the dominance of the goal value over the reward value model observed in Studies 1–2 reflected the hypotheticality of the choices in these studies, re-directing attention towards the goal and away from potential rewards. While this would not straightforwardly account for overall value effects observed in previous studies involving hypothetical choice[36], we sought to rule this possibility out as

**Table 1 Comparison of overall reward vs overall goal value effects on log RT**

| Predictors | Study 1 | | | | | Study 2 | | | | |
|---|---|---|---|---|---|---|---|---|---|---|
| | Estimate | CI | t | df | p | Estimate | CI | t | df | p |
| *OV_reward by Choice Goal* | | | | | | | | | | |
| (Intercept) | 1.42 | 1.36–1.49 | 41.93 | 31 | **<0.001** | 1.52 | 1.44–1.59 | 38.93 | 31 | **<0.001** |
| Value Difference | −0.49 | −0.55–−0.43 | −16.48 | 3430 | **<0.001** | −0.43 | −0.52–−0.34 | −9.18 | 31 | **<0.001** |
| Overall Value | 0.05 | −0.01–0.11 | 1.61 | 3448 | 0.108 | −0.01 | −0.06–0.05 | −0.24 | 4222 | 0.809 |
| Best - Worst Condition | −0.07 | −0.14–0.00 | −1.90 | 31 | 0.066 | −0.01 | −0.09–0.06 | −0.33 | 31 | 0.740 |
| Overall Value: Best - Worst | −0.75 | −0.86–−0.63 | −12.66 | 3296 | **<0.001** | −0.76 | −0.86–−0.65 | −14.39 | 3914 | **<0.001** |
| *OV_goal + OV_reward* | | | | | | | | | | |
| (Intercept) | 1.42 | 1.36–1.49 | 41.97 | 31 | **<0.001** | 1.52 | 1.44–1.59 | 38.93 | 31 | **<0.001** |
| Value Difference | −0.49 | −0.55–−0.43 | −16.48 | 3430 | **<0.001** | −0.43 | −0.52–−0.34 | −9.18 | 31 | **<0.001** |
| Overall Goal Value | −0.37 | −0.43–−0.32 | −12.66 | 3296 | **<0.001** | −0.38 | −0.43–−0.33 | −14.39 | 3914 | **<0.001** |
| Overall Reward Value | 0.05 | −0.01–0.11 | 1.61 | 3448 | 0.108 | −0.01 | −0.06–0.05 | −0.24 | 4222 | 0.809 |
| Best - Worst Condition | −0.08 | −0.15–−0.01 | −2.21 | 31 | **0.035** | −0.03 | −0.11–0.04 | −0.85 | 31 | 0.402 |

Significant effects are highlighted in bold.

**Table 2 Model comparison for OV_reward and OV_goal effects on RT across studies**

| Model | Study 1 | | | | | Study 2 | | | | |
|---|---|---|---|---|---|---|---|---|---|---|
| | $R^2$ | AIC | dAIC | $X^2$ | p | $R^2$ | AIC | dAIC | $X^2$ | p |
| Model 0 (baseline): VD + $C_{b-w}$ | 0.22 | 4112 | | | | 0.19 | 4396 | | | |
| Model 1: baseline + $OV_{reward}$ | 0.22 | 4112 | 0 | 2.72 | 0.099 | 0.19 | 4398 | 2 | 0.04 | 0.847 |
| Model 2: baseline + $C_{b-w}$: $OV_{reward}$ | 0.26 | 3956 | −156 | 157.68 | **<0.001** | 0.25 | 4197 | −202 | 203.31 | **<.001** |
| Model 3: baseline + $OV_{goal}$ | 0.26 | 3957 | 1 | 2.59 | 0.108 | 0.25 | 4195 | −2 | 0.06 | 0.809 |

For each study, models are compared sequentially, and dAIC is the difference in AIC of each model to the previous model. *VD* Value Difference, *OV* overall value, significant effects are highlighted in bold

well by performing a follow-up study in which choices were incentivized rather than hypothetical, with participants receiving one of the products from the study based on their indication that it was the most or least preferred item in a set (Supplementary Study 2; see Supplementary Methods). This study replicated the findings observed across Studies 1–2 (Supplementary Discussion 2); critically, RTs were still not influenced by overall reward value ($p = .838$) and continued to be best accounted for by the goal value models (dAICs $= −18$ to $−16$).

**Goal value LCA explains behavioral effects of overall value.** Previous research has shown that the dynamics of value-based decision-making can be captured by models of accumulating evidence to bound[9,10,46,47]. This work has demonstrated that a subset of such models can capture the speeding effects of both overall reward value and value difference on choice RTs. For instance, these effects emerge naturally from a model in which option values accumulate additively and compete with one another (the leaky competing accumulator [LCA] model[48]). However, previous implementations of this and related models of value-based decision-making have assumed that activity in the model is driven by the reward value of one's options, and therefore predict that overall value will only ever result in speeding of choice RTs. Accordingly, when we simulate such a model we find that it replicates behavior on our Choose-Best task (consistent with past work) but not our Choose-Worst task (Supplementary Fig. 1).

To account for our findings, we developed a modified LCA model. Rather than accumulating evidence of potential rewards, this model accumulates evidence in favor of one's task goal (i.e., goal value). According to this model, higher reward associated with a given option would constitute strong evidence in favor of one's goal during the Choose-Best task but weak evidence in favor of one's goal in the Choose-Worst task. Choice behavior simulated by this model qualitatively replicates all of our key findings across Studies 1 and 2: (1) the effect of overall reward value on choice RTs reverses between Choose-Best trials (speeding) and Choose-Worst trials (slowing) (Fig. 1d); (2) overall goal value has a consistent speeding effect, and is sufficient to account for RTs across both tasks (Supplementary Fig. 2); (3)

value difference has a consistent speeding effect across the two tasks (Supplementary Fig. 3); and (4) overall choice RTs are similar across the two tasks (Supplementary Fig. 3).

**Value network differentially tracks reward & goal congruency.** Previous work has identified a network of brain regions – primarily comprising ventromedial prefrontal cortex (vmPFC) and ventral striatum – that tracks both the overall reward value of a choice set and the relative reward value of the chosen item (the signed difference between the reward associated with the chosen vs. unchosen items)[15,31,35,36]. However, because these findings are based entirely on tasks that require participants to choose the best option, it is unclear whether neural activity is determined by the reward value of those items (as implied by past studies) or by the goal congruency of those reward values. To disentangle these two possibilities, we had participants in Study 2 perform our Choose-Best and Choose-Worst tasks while undergoing fMRI. The fact that reward and goal congruency were orthogonalized by design (Supplementary Fig. 4) allowed us to compare the degree to which activity in a previously defined valuation network[15] was associated with overall and relative (chosen minus unchosen reward value vs. chosen minus unchosen goal value) reward values and goal values.

To test the hypotheses that the value network tracks reward value (GLM-1) or goal congruency (GLM-2), we performed separate regressions correlating single-trial BOLD activity in this network with either relative and overall reward value (GLM-1) or relative and overall goal value variables (GLM-2). We found that the goal value model (GLM-2) provided a significantly better fit to neural activity ($X^2 = 5.67$, $p < 2e{-}16$; Table 3), with significant effects of both overall ($b = 0.16$, $t = 2.45$, $p = .014$) and relative goal value ($b = 0.23$, $t = 3.50$, $p < .001$). Both goal value variables continue to be significant when including reward-related variables in the same model (GLM-3; Table 4; Fig. 2). Interestingly, in addition to tracking overall and relative goal value, the value network also tracked overall reward value ($b = 0.21$, $t = 3.17$, $p = .001$; Table 4). In other words, this network tracked overall and relative value of a choice set in terms of their goal congruence, while also tracking the overall reward associated with those options (irrespective of the choice goal). None of our GLMs

---

**Table 3 Model comparison for reward and goal value – related BOLD activity in the valuation network ROI**

| Model | $R^2$ | AIC | BIC | dAIC | Chi$^2$ | p |
|---|---|---|---|---|---|---|
| $C_{b\text{-}w}$ + RT (baseline) | 0.062 | 11976 | 12040 | | | |
| GLM-1: baseline + RV$_{reward}$ + OV$_{reward}$ | 0.066 | 11969 | 12045 | −7 | 11 | **0.004** |
| GLM-2: baseline + RV$_{goal}$ + OV$_{goal}$ | 0.065 | 11963 | 12040 | −6 | 5.68 | **<0.001** |
| baseline + RV$_{goal}$ + OV$_{goal}$ + OV$_{reward}$ | 0.069 | 11955 | 12038 | −8 | 10.2 | **0.001** |
| GLM-3: baseline + RV$_{goal}$ + OV$_{goal}$ + RV$_{reward}$ + OV$_{reward}$ | 0.069 | 11957 | 12046 | 2 | 0.2 | 0.657 |

RV Relative Value, OV = Overall Value, significant effects are highlighted in bold

---

**Table 4 Fixed effects summary GLM-3: Reward and goal values**

| Predictors | Estimates | CI | t | df | p |
|---|---|---|---|---|---|
| (Intercept) | 0.00 | −0.08–0.08 | 0.01 | 31.00 | 0.992 |
| Best - Worst Condition | −0.01 | −0.10–0.09 | −0.13 | 42.00 | 0.896 |
| Overall Reward Value | 0.21 | 0.08–0.34 | 3.15 | 3844.00 | **0.002** |
| Relative Reward Value | −0.03 | −0.16–0.10 | −0.44 | 4225.00 | 0.657 |
| Overall Goal Value | 0.16 | 0.03–0.29 | 2.44 | 1632.00 | **0.015** |
| Relative Goal Value | 0.22 | 0.09–0.36 | 3.37 | 4173.00 | **0.001** |
| RT | 0.02 | −0.08–0.12 | 0.37 | 32.00 | 0.715 |

Significant effects are highlighted in bold

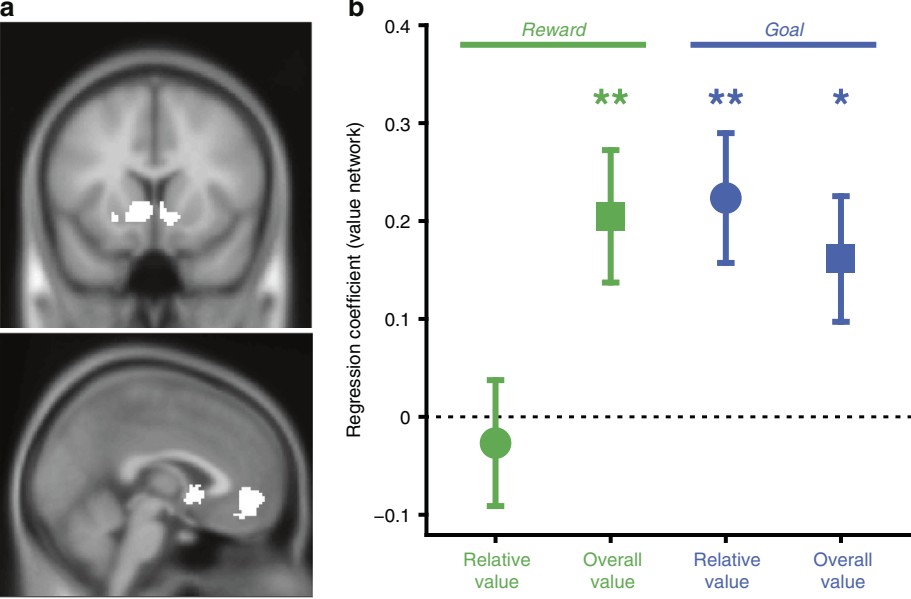

**Fig. 2** The valuation network tracks goal values and overall, but not relative reward value. **a** Valuation network mask. **b** Mixed-effects regression coefficients show that the valuation network ROI defined a priori based on ref. [15]. tracks both overall and relative goal value, and also tracks overall (but not relative) reward value. Error bars show standard error of the mean. *$p < 0.05$, **$p < 0.01$

revealed a significant correlation between BOLD activity and relative reward value (i.e., chosen vs. unchosen reward).

This general pattern held across the two constituent regions of the valuation network (ventral striatum and vmPFC, Supplementary Fig. 5, Supplementary Table 3), as well as posterior cingulate cortex (PCC, Supplementary Fig. 6, Supplementary Table 4), a region that often coactivates with the valuation network. The only discernible difference between these regions was that vStr and PCC appeared to be more weakly associated with overall goal value than vmPFC was.

**Reward vs. goal congruency spatially dissociated in striatum.** Since we failed to observe relative reward value signals in our a priori ROI, we performed a follow-up whole-brain analysis (mirroring GLM-3) to test whether a subset of this network, or regions outside of it, track this variable. We did not find such correlates, even at a liberal threshold (voxelwise uncorrected $p < 0.01$, $k \geq 10$ voxels). However, our whole-brain analysis did reveal an unexpected functional dissociation within the striatum, with more dorsal regions primarily tracking overall reward value and more ventral regions primarily tracking relative goal value (Fig. 3a). To test this dorsal-ventral dissociation directly, we defined ROIs for three sub-regions along a dorsal-ventral axis previously shown to seed distinct resting-state networks Fig. 3b[49]. We found a significant interaction between value variables and striatal sub-regions ($F_{(4, 25560)} = 2.81$, $p = 0.024$), such that sensitivity to overall reward value was greatest in the dorsalmost sub-region (dorsal caudate; $b = -0.21$, $t = -2.10$, $p = 0.036$) whereas sensitivity to relative goal value was greatest in the ventralmost sub-region (inferior ventral striatum; $b = 0.24$, $t = 2.19$, $p = 0.029$, Fig. 3c, Supplementary Table 5).

Unlike in striatum, an additional exploratory test of analogous differences across subregions in vmPFC (rostral ACC vs. medial orbitofrontal cortex[36]) did not reveal significant interactions of any of our variables with subregion (Supplementary Table 6), though trends were observed differentiating rACC's tracking of overall reward value and from mOFC's tracking of overall and relative goal value (Supplementary Table 7).

**Common correlates of overall reward value and set appraisal.** We have previously shown that the overall reward value of an option set predicts how attractive a participant will appraise that set to be (set liking), and that these liking ratings correlate with value network activity[14,36]. Set liking correlated with activity in these regions irrespective of whether participants were instructed to appraise the set on a given trial, suggesting that these appraisals are triggered relatively automatically[32,36,50]. Unlike previous work, our current studies were able to distinguish whether set liking was driven by overall reward value or overall goal value, and to compare the neural correlates of set liking with the neural correlates of both types of overall and relative value. Across both studies, we found that set liking correlated with overall reward value (Study 1: $b = 6.35$, $z = 25.44$, $p < 0.001$, Study 2: $b = 4.87$, $z = 20.74$, $p < .001$) and not overall goal value (Study 1: $b = -0.07$, $z = -0.53$, $p = 0.599$, Study 2: $b = 0.09$, $z = 0.68$, $p = 0.497$). Consistent with previous findings, in Study 2 we found that activity in the value network was correlated with set liking ($b = 0.16$, $t = 2.22$, $p = 0.027$). While set liking correlates did not differ significantly across regions of striatum ($F_{(2, 12295)} = 0.72$, $p = 0.485$), it is notable that the qualitative pattern of responses (numerically higher in the more dorsal regions) most closely matched the pattern we observed for overall reward value (Supplementary Fig. 7). Together these findings provide tentative but convergent evidence that overall reward value signals may be related to a process of automatic appraisal.

**Discussion**
In spite of the heterogeneity of approaches in research on value-based decision-making, a consistent set of findings has emerged with respect to the behavioral and neural correlates of choice values. However, up to this point it has been unknown whether these correlates reflect the reward value of those options (i.e., how much reward one can expect for obtaining a given option) or their goal congruency (i.e., to what degree does a given option support one's current choice goal). To disentangle these accounts of behavioral and neural data, we had participants make value-based choices under two different choice goals, either to choose the most rewarding or least rewarding option. We found that

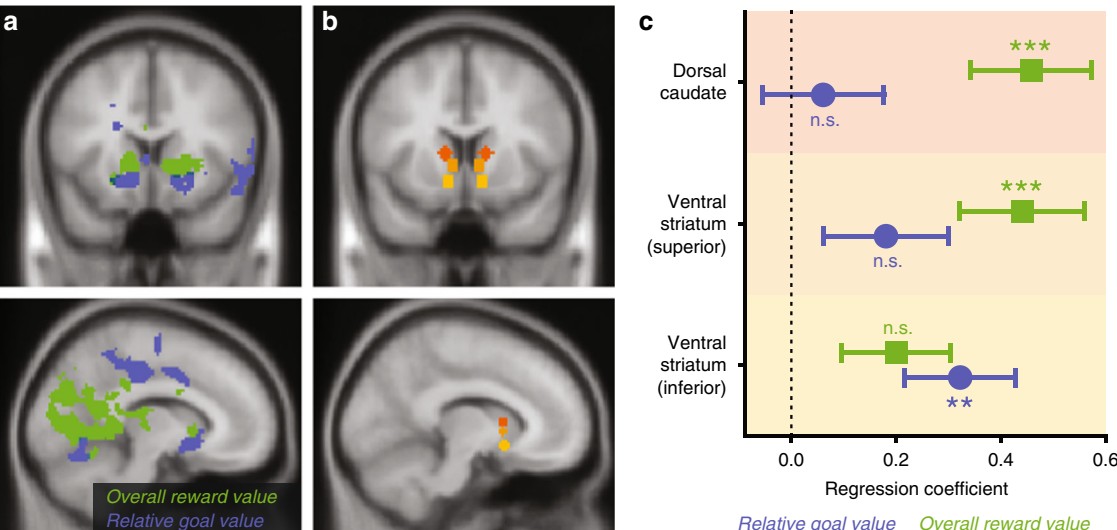

**Fig. 3** Reward and goal value dissociate across the striatum's dorsal-ventral axis. **a** Whole brain results for relative goal value (blue) and overall reward value (green), thresholded at voxelwise $p < 0.001$ and cluster-corrected $p < 0.05$. Despite being correlated with activity in our a priori value network ROI, overall goal value did not survive the whole-brain threshold used for these follow-up exploratory analyses. **b** To interrogate our findings across regions of striatum, we selected independent bilateral ROIs previously used as seeds for distinct resting-state networks:[49] Dorsal Caudate (dark orange; [x, y, z] = ±12, 10, 8), Ventral Striatum, superior (orange; ±8, 10, 1), and Ventral Striatum, inferior (yellow; ±10, 11, −9). **c** Mixed-effects regression coefficients across these ROIs demonstrate a dorsal-ventral dissociation, with dorsal regions more sensitive to overall reward value and the ventralmost region more sensitive to relative goal value. Error bars show standard error of the mean. $^{**}p < 0.01$, $^{***}p < 0.001$

choice RTs and neural activity in value-related regions were accounted for by overall and relative goal value (increasing with the congruency of option values with the current choice goal). We additionally identified neural (but not behavioral) signatures of overall reward value. Together, these results provide a cognitive, rather than reward-centered explanation of hallmark findings in value-based decision-making, and call for a significant reinterpretation of past research.

Previous studies on value-based decision-making have consistently found that participants make faster decisions as the overall reward value of their options increases[11–14,51]. It has been natural to assume that these speeding effects are related to the rewarding properties of these choice options, particularly given that approach is known to be facilitated by reward and impeded by punishment[26,27] and response vigor is known to increase in high reward rate environments[28–30]. However, these studies confounded reward value and goal congruency, concealing an alternative, cognitive explanation of these findings, that speeding effects of overall value reflect increased activation of responses specific to one's goal[10,45,48,52]. Our findings support the latter account, demonstrating that a choice set's overall reward value speeds choices only when the participant's goal is to choose the most rewarding item, and that it has the opposite effect when the goal is to choose the least rewarding item.

Our neural findings serve to similarly qualify reward-based accounts of neural data. Previous neuroimaging research has linked activity in a common set of brain regions to the relative and overall reward value of a choice set. On their face, those findings could be interpreted as reflecting a directionally consistent coding of potential rewards[31,37,42], such that (a) one's preference for a given item could be read out from activity in these regions and (b) the preferences coded in these regions could feed directly into a choice between the choice options (reflected in relative value signals). On such an account, relative value signals should always reflect whether the chosen option is generally more or less preferred than the unchosen option(s), and overall value should always reflect how much those options are generally preferred. This form of directionally consistent coding scheme is

intuitive and can be seen as evolutionarily adaptive because it enables an organism to rapidly determine which options are most approach-worthy. Critically, this value code is also sufficient to perform our tasks – providing participants just as much information about the least preferred items as it does the most preferred – obviating the need for any additional, recoded value signals[53]. We found neural signals that are inconsistent with this coding scheme, but instead tracked options' goal congruency. These overall and relative goal congruency signals would not be predicted by a reward coding account but, together with our behavioral and modeling results, are consistent with an alternate account whereby activity in these valuation networks reflects the accumulation of goal-relevant information leading up to a choice[10,13], including information about potential rewards and their relationship to one's current task goal[54].

In addition to these neural signatures of goal congruency, we also observed neural signals that tracked the overall value of a choice set in a goal-independent (i.e., directionally consistent) manner. That is, the value network also tracks how much reward can be expected from the choice set, across both Choose-Best and Choose-Worst tasks. Notably, unlike our goal-dependent signals, these goal-independent overall value signals were not accompanied by goal-independent relative value signals (i.e., chosen minus unchosen reward), as would be expected if this circuit was comparing items according to their reward value. It may therefore be parsimonious to interpret these overall reward signals as reflecting an appraisal process that occurs in parallel with the choice process[14,36], cf. [55], or preceding it[56]; rather than reflecting a component of the decision-making process[36]. With that said, we emphasize that these inferences pertain specifically to BOLD signals within vmPFC and ventral striatum, and that it is entirely possible that other coding schemes would be found at the single-unit level within these or other regions.

There is another potential explanation for the overall reward value and relative goal value signals we found juxtaposed in the value network, which focuses on the affective qualities these variables share[36]. Specifically, we and others have shown that overall reward value predicts how positively a person will feel

about a set of items[14,36] and that the value network tracks such affective appraisals whether or not they are task-relevant[32,36]. Similarly, relative goal value is a proxy for confidence in one's decision, as well as for the ease or fluency of a task. Both of these engender positive affect[57–60] and both have been linked with activity in the value network[36,61–65]. This shared association with affect offers a salient (but non mutually exclusive) alternative to accounts of value network findings that focus on decision-making and other cognitive processes. However, further research on this circuit is necessary to tease apart affective and cognitive contributions to neural activity.

Within striatum, we found a surprising anatomical dissociation, whereby dorsomedial vs. ventral regions were differentially associated with overall reward value and relative goal value. This dissociation was most strongly driven by an effect of overall reward value within dorsal caudate that was absent for relative goal value (whereas both variables were associated, to different degrees, with activity in ventral striatum). While not predicted, we speculate that this pattern may reflect the differential relationship of the two variables with affective and motivational states. Specifically, overall reward value is not only associated with affective appraisal (as noted above), but can also serve to motivate approach behavior. Accordingly, previous findings have linked dorsomedial striatum with action valuation[66–68] and motivation[69–71]. It is also noteworthy that in spite of their relatively close spatial proximity, these ROIs fall within separate networks with distinct patterns of resting-state functional connectivity – the ventralmost region is associated with the limbic network and the dorsalmost region is associated with the frontoparietal control network[49,72,73]. Therefore, an alternative account of this dorsal caudate finding is that overall reward value, being a task-irrelevant variable (cf. distractor), may elicit frontoparietal activity because of the increase in control demands. This interpretation is consistent with a similar dissociation recently observed by Fischer and colleagues[74]. They had participants perform a learning task and found that ventral striatum tracked short-term rewards whereas dorsal striatum generally tracked long term rewards, which had to be inferred from the task environment[74]. However, they found that dorsal striatum also tracked short-term rewards and the extent to which it did so predicted better learning of long-term rewards, which these authors interpreted as reflecting increased control that was triggered by this region's encoding of salient information (short-term reward) that was irrelevant to the task of inferring long-term reward. In spite of this converging evidence, we emphasize that the interpretations of our striatal findings are post hoc and therefore should be taken with caution. Nevertheless, they suggest avenues for further inquiry that could shed light on this intriguing finding.

Our findings have important implications for future research on value-based decision-making within healthy and clinical populations. We show that traditional approaches to studying such decisions conflate two types of variables: those that are specific to rewards per se and those that are related to how consistent rewards are with choice goals. We prescribe a method for pulling these two types of signals apart, and show that doing so reveals distinct behavioral and neural patterns. This method could, for instance, be employed to identify mechanisms that are shared (e.g., goal-related) and distinct (e.g., outcome-related) when choosing among appetitive options vs. aversive options[25,75].

In addition to informing basic research into distinct processes supporting value based choice (reward processing vs. goal-driven behavior), our work carries important implications for research on clinical populations that exhibit aberrant reward processing[70,71,76–78] or impairments in translating rewards into goal-driven behavior[79]. For instance, to the extent these individuals exhibit impulsive behaviors linked to reward rather than goal values, we provide an approach for uniquely identifying this variable of interest and honing in on the relevant deficits. By decoupling these two processes, future research can identify the best approaches for treating the relevant deficits, or at least rule out the worst ones.

## Method

**Participants**. Participants were recruited from Brown University and the general community. For Study 1, 37 participants were recruited. Of these, seven were excluded, one due to previous participation in a similar experiment, three due to incomplete sessions, and three due to insufficient variance in product evaluation, precluding the generation of sufficient choice sets. Thus, 30 participants (76.7% female, $M_{age}$ = 20.3, $SD_{age}$ = 2.1) were included in our analyses. For Study 2, 31 participants were recruited and one was excluded due to an incomplete session. Thus, 30 participants (56.67% female, $M_{age}$ = 21.87, $SD_{age}$ = 4.40) were included in our analyses. Participants in all studies gave informed consent in accordance with the policies of the Brown University Institutional Review Board.

**Procedure**. The experiment was computerized using Matlab's Psychophysics Toolbox[80,81]. Participants first evaluated a series of products according to how much they would like to have each one. Items were presented individually, and participants used a mouse to indicate their evaluation of that item along a visual analog scale, anchored at 0 ("not at all") and 10 ("a great deal"). Participants were encouraged to use the entire range of the scale and reserve 0 and 10 for extreme values. Based on these subjective evaluations choice sets were created, such that half of the choices primarily varied in relative value (RV; e.g., best vs. average of the remaining items), while the other half primarily varied in overall value (OV; average value of items in a set). Additional details regarding distribution of items and process of choice set construction can be found in[82]. Participants then viewed sets of four products and were instructed either to choose the item they most preferred Choose-Best condition; cf[14,36,82]. or to choose the item they least preferred (Choose-Worst condition). The different choice goals (Choose-Best vs. Choose-Worst) were performed in separate blocks, and the order of the two blocks was counterbalanced across participants. Stimuli remained on the screen until a choice was made. In Study 1, participants performed a total of 120 choices, 60 for each condition. They indicated their choice with a mouse click. There were no response deadline in the initial rating or choice. The general procedure in Study 2 was identical to that in Study 1 except that (1) choices were performed in the scanner; (2) they completed 144 choices (72 for each choice goal); (3) the interval between choices was varied across trials (2–7s, uniformly distributed); and (4) choices were made with a button press rather than with a mouse. Participants responded with the index and middle fingers of their left hand and right hand, using an MR-safe response keypad. Response keys were mapped to locations on the screen (upper left, lower left, upper right and lower right). Participants practiced the response mappings prior to the choice phase by pressing the button corresponding to a given cued location until they gave the correct response on 15 consecutive trials.

**fMRI data acquisition and analysis**. *Imaging Parameters*: Scans were acquired on a Siemens 3T PRISMA scanner with a 64-channel phase-arrayed head coil, using the following gradient-echo planar imaging (EPI) sequence parameters: repetition time (TR) = 2500 ms; echo time (TE) = 30 ms; flip angle (FA) = 90°; 3 mm voxels; no gap between slices; field of view (FOV): 192 × 192; interleaved acquisition; 39 slices. To reduce signal dropout in regions of interest, we used a rotated slice prescription (30° relative to AC/PC). The slice prescription encompassed all ventral cortical structures but in a few participants omitted regions of dorsal posterior parietal cortex. Structural data were collected with T1-weighted multi-echo magnetization prepared rapid acquisition gradient echo image (MEMPRAGE) sequences using the following parameters: TR = 2530 ms; TE = 1.69 ms; FA = 7°; 1.0 mm isotropic voxels; FOV = 256 × 256. Head motion was restricted with a pillow and padding. Stimuli were generated using Matlab's Psychophysics Toolbox (Matlab 2013a) presented on a 24″ BOLDscreen flat-panel display device (Cambridge Research Systems) and were viewed through a mirror mounted on the head coil.

*fMRI Analysis*: fMRI data were processed using SPM12 (Wellcome Department of Imaging Neuroscience, Institute of Neurology, London, UK). Raw volumes were realigned within participants, resampled to 2 mm isotropic voxels, non-linearly transformed to align with a canonical T2 template and spatially smoothed with a 6 mm full-width at half-max (FWHM) Gaussian kernel.

*Trial-wise ROI analyses*: Preprocessed data were submitted to linear mixed-effects analyses using a two-step procedure. In the first step, we computed first-level general linear models (GLM) in SPM to generate BOLD signal change estimates for each trial and participant. GLMs modeled stick functions at the onset of each trial. Trials were concatenated across the two task blocks and additional regressors were included to model within-block means and linear trends. GLMs were estimated using a reweighted least squares approach RobustWLS Toolbox;[83] to minimize the influence of outlier time-points (e.g., due to motion). The obtained

estimates were transformed with the hyperbolic arcsine function (to achieve normality), and then analyzed using LMMs using lme4[84] in R[85].

Of primary interest in the present study is activation in the valuation network, which was obtained using the conjunction mask from Bartra, et al.[15]. These results were followed up with separate masks for the vmPFC and vStr clusters within this ROI. PCC activity was extracted from an ROI identified in a previous study[14].

To test for gradients within vStr, we constructed spheres ($r = 4$ mm) around seeds in three regions along the caudate's dorsal-ventral axis most strongly associated with the frontoparietal control network (left hemi coordinates: $-12, 10, 8$), the default mode network ($-8, 10, 1$), and the limbic network, respectively ($-10, 11, -9$)[49]. For the gradient analysis of these spheres, the data were best fit modeling linear as well as quadratic effects for region. ROIs for rACC and mOFC to test for differences within vmPFC were selected anatomically, analogous to a previous study[36].

*Whole-brain analyses*: We complemented the ROI analyses with whole-brain GLMs. For these analyses, we computed first-level GLMs, modeling stick function at stimulus onsets, and parametric regressors for (1) choice goal, (2) reward-related OV, (3) goal-related OV, (4) reward-related chosen vs. unchosen value and (5) goal-related chosen vs. unchosen value. Regressors were de-orthogonalized to let them compete for variance. As above, trials were concatenated across the two task blocks, additional regressors were included to model within-block means and linear trends, and GLMs were estimated using RobustWLS. Second level random effects analyses on first-level estimates were performed using SPM with voxel-wise thresholds of $p < 0.001$ and cluster-corrected thresholds of $p < 0.05$. Results were visualized using XJview (http://www.alivelearn.net/xjview).

*Analysis*: Scripts for all analyses are available under https://github.com/froemero/goal-congruency-dominates-reward. Data were analyzed using lme4 package[84] for R version 3.4.3[85]. RTs were analyzed with linear mixed effects models (LMMs) and choices were analyzed with generalized linear mixed effects models (GLMMs) with a binomial link function. When degrees of freedom are provided, these are Kenward Roger approximated and underlie corresponding $p$-values (both implemented in the sjplotpackage for R). Else, we used the lmerTest package to obtain $p$-values based on Satterthwaite approximation of degrees of freedom. Choice consistency was determined based on whether participants chose the item with the highest value in Choose-Best, and conversely the lowest value in Choose-Worst. Analyses of choice consistency tested the probability of choosing the option that best satisfies the current choice goal (lowest-valued item for Choose-Best, highest-valued item for Choose-Worst); these analyses exclude trials for which multiple options equally satisfy these instructions (i.e., when multiple options share the lowest or highest value). Response times were log-transformed to reduce skew. Value difference and overall value were mean-centered and rescaled to a range from zero to one prior to analysis. Choice condition (Choose-Best vs. Choose-Worst) was included as a factor to test for main effects of choice goal and interactions between goal and choice values. We used a sliding difference contrast that estimates the difference in means between subsequent factor levels with the intercept (baseline value) being estimated as the mean across both conditions. Random effects were specified as supported by the data, according to singular value decomposition[86,87].

*Computational model*: To directly compare reward-based and goal-based accounts of our findings, we simulated data using two versions of a leaky competing accumulator model LCA;[48]. In an LCA model, evidence for each potential response (in our case, each of four choice options) accumulates noisily over time based on its respective input ($I_{Option}$). As the evidence for each response accumulates, it competes with the evidence accumulated for each of the other responses, according to a mutual inhibition term ($w$). Over time, accumulated evidence decays according to a leak (or forgetting) term ($k$). A decision is made once activation of one of the options reaches a decision boundary ($z$), and the time at which this threshold is crossed is recorded as the decision time. This decision time is combined with a fixed non-decision time (reflecting stimulus processing and response execution; $t_0$) to generate the response time for a given model simulation. Unlike the standard drift diffusion model, the LCA has additive properties that produce RTs that vary with both the relative and overall value of its inputs[48].

We compared two versions of the LCA model to see which could better account for our findings. One version used reward values as inputs as in previous studies[88]; while a second, modified form of this model used goal values as inputs. In the Choose Best condition, goal values are identical to reward values. For the Choose Worst condition goal values were computed by recoding reward values, such that lower reward values had higher goal value and vice versa. Both models otherwise used identical parameters, which we determined by performing gradient descent (using *fmincon*) to maximize the likelihood of Study 1 data (choices and RTs) given model predictions, collapsing across participants. We generated trial-wise predictions by simulating the LCA's accumulation-to-bound process 1000 times for each unique choice set in the experiment. The best-fit model parameters were as follows: z = 2.243, s(noise coefficient) = 0.587, k = 0.153, w = 0.671, $t_0$ = 0.849 s (log likelihood = $-12883.128$). The goal value model simulations reported in the main text use these parameter values but all of the relevant qualitative predictions of these models are robust to variation in these parameters. We did not perform additional model fits or simulations for an analogous reward value-based LCA model because it is necessarily unable to incorporate information about the different choice goals and will hence inevitably always choose the best option

regardless of the task. Model simulation and parameter estimation were performed using custom Matlab (2017b) scripts.

*Visualization*: Figures displaying projected values and confidence intervals of response times were generated using effects and ggplot2 packages based on the relevant LMMs. For viewing purposes, centered and scaled predictor variables are rescaled to their original values prior to plotting.

**Reporting summary**. Further information on research design is available in the Nature Research Reporting Summary linked to this article.

## Data availability
The datasets generated and analyzed during the current study are available from the corresponding author on reasonable request.

## Code availability
Scripts for all analyses are available under https://github.com/froemero/goal-congruency-dominates-reward.

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

## Acknowledgements
This work was funded by a Center of Biomedical Research Excellence grant P20GM103645 from the National Institute of General Medical Sciences. The authors are grateful to Ayenna Cagaanan, Michelle Basta, Hattie Xu and Maisy Tarlow for assistance in data collection, and to Wil Cunningham, Scott Guerin, Uma Karmarkar, Matt Nassar, and Avinash Vaidya for helpful comments.

## Author contributions
A.S. and R.F. conceived the study. R.F. and C.D.W. collected the fMRI data. R.F. analyzed the data. R.F. and A.S. interpreted the results. All authors contributed to programming the tasks. A.S. and R.F. wrote the manuscript, and all authors edited the manuscript.

## Competing interests
The authors declare no competing interests.
