## [Peer Review File · Nature Communications]

Reviewers' Comments:

Reviewer #1:

Remarks to the Author:

This manuscript reports a series of behavioral studies, combined with fMRI and computational modeling, that addresses value representation in decision making. The task was designed to disentangle the reward value of the presented stimuli from the goal (value). The authors report that relative goal value drives behavior and is represented in an a priori defined valuation network (vmPFC and striatum). In addition, while having no effect on behavior, the overall reward value of the presented stimuli is reflected in the striatum, mainly in the dorsal caudate. The authors argue that unlike in previous studies addressing valuation, goal value and stimulus (reward) value need to be deconfounded.

This is a well-designed and thoroughly performed and analyzed study. The methods are sound and state of the art and the inferences made by the authors are covered by the data. However, I am not sure how novel and surprising these results are and whether they signify an important advancement of our knowledge. In contrast to habitual behavior (which is not studied here), goal-directed behavior such as value-based or perceptual decision making is driven by the goals of the participant. I would not have expected that general value of a stimulus which does not have any relevance for the outcome of the task at hand but only serves as a feature dimension (similar as color, motion direction etc. in perceptual decision making) based on which a decision has to be made has a major influence on behavior. For participants it was irrelevant, whether they would like to have a stimulus or not, because the motivation and goals of participating in the task were completely uncoupled from the stimuli. It does not seem to matter, whether I have to make a judgment who of my colleagues wears the most or least expensive wearables or whether I discriminate the color of these wearables. "Reward value" might not be the best-suited term here: what the authors seem to mean is the value that participants would assign each item, had they the opportunity to get one of them. As the authors state themselves, reward value is -at least motivationally- irrelevant for the participants in the task. The actual reward participants expect from participating consists of many outcomes, including course credits/payment, doing this task at a minimal time, being perceived by the experimenter as a cooperative and intelligent subject etc. These rewards determine the (sub)goal of making the choices according to the instructions, which in turn drives decision making and behavior. The entire issue gets even clearer, when one imagines to transfer the task to a monkey lab. Monkeys would quickly learn that the values the associate with shown items (e.g., many grapes vs. one grape) are completely motivationally irrelevant, if their choice is rewarded only with a drop of water. Their behavior would not be influenced by pictures of rewarding items any more.

To me, the only surprising finding is that the overall reward value (i.e. the values they would assign to the items if they could get them, which, of course, is closely related to the set liking assessed in the post-trial survey) is also represented in the dorsomedial striatum. Of course, to do the task, the values of the items have to be retrieved, such that their representation in vmPFC could be expected, but the fact that the striatum, a structure related to learning and decision making, also represents them may be less expected.

Could it be possible that the representation of the overall reward value was actually related to the anticipation of the upcoming post-choice affective rating of set appraisal? Related to this, the exact timing of stimulus presentation and choices should be reported in more detail (at least in the supplemental information).

Generally, the authors may be correct that in previous studies many researchers have overlooked the potential difference between values of an item and the actual outcome of an action (i.e. goal achievement). So, perhaps it might be a good idea to clarify this point which every good experimenter should always keep in mind. It is generally important to be as clear as possible with the terms used in the manuscript. Can one really talk about "reward value" and "preferred" option in the present task in the strict sense?

The striatal gradient reported for the overall reward value is interesting. I wonder whether and -if so- how this fits to a recent study in which a striatal gradient in model updating based on simple reinforcement learning vs. long-term optimal inference was found (Fischer, Bourgeois-Gironde, Ullsperger, Nat Commun, 2017). There, participants experienced incongruency of (long-term) goal and short-term reward. Optimal value updates guiding their upcoming choices in the service of the long-term goal were represented most strongly in mediodorsal striatum (caudate) whereas the (short-term) reward prediction error biasing their choice towards suboptimal behavior was most strongly represented in the ventral striatum. Can this be reconciled with the present results?

Further points:

Please provide more detail about the ROI. A figure showing the extent of the vmPFC and striatal regions would be quite informative to the readers, particularly those who are less familiar with the field.

I would also find it interesting to see the same subregional analyses provided for the striatum done for the vmPFC.

Please provide more details on the fitting of the LCA models. What exactly is meant by "manual fitting"?

Very minor:

p. 9, abbreviation VD was not introduced beforehand

SI, p. 2 l. 28: doubled "were recruited"

Reviewer #2:

Remarks to the Author:

Frömer et al. report the separation of reward and goal-consistent influences on choice. To separate the effects of task-goal value and overall reward value, the authors use a task in which participants are explicitly instructed to choose either the worst or best option. When compared, choices in the choose-worst task are described as inconsistent with overall reward while consistent with goal choices. Reward value should be the only predictor of choice and response time If choice is only driven by the maximization of the utility of goods received (or considered). If task-goal utility drives reward experiences then response times and brain activation in the reward system should reflect an interaction with task goal. The authors find that response time is best predicted by task-goal congruence and that brain reward system reflects a combination of both as well as value differences between options. They conclude that the previous research on value-based choice should be reinterpreted in light of this finding.

The study is designed to address an interesting question: Is there a reward system in the brain that is removed from the process of motivating goals and carrying out tasks? The authors employ an interesting task that provides a useful way to directly contrast traditional definitions of reward and task-related reward. I view the research as making an interesting contribution to the conversation of over how choice is modeled that is best framed in the domain of contextual and non-traditional components of a utility model.

1. The overall tone of the paper is one of redefining what reward is and whether it is task-defined or more generally part of an approach-avoid system (described as Pavlovian in the text). The paper makes mixed use of different descriptions of utility, context, and traditional approach system distinctions (Pavlovian vs. Operant) while not fully addressing the literature in any of these domains, making it very challenging to understand the implications of the work or what exactly should be reinterpreted. The best option moving forward would seem to be to focus on context

effects on value (see work by Rangel; Lee and Sul, Neuron; Clithero et al Choice independent value etc.). The manipulation from this paper could be seen as another contributor to a multifaceted utility function through which context, local goals, and other factors contribute to option selection.

2. Relative weights of task and general rewards. The paper works to emphasize the role of task-goals but there are several potential issues: A. Hypothetical choices reduce the magnitude of representation expected from the choice options independent of goal. B. RT model comparison analysis shows borderline marginal effect of reward. It is disingenuous to describe this as showing effects exclusive to task even if the effects is not (strictly speaking) significant. This is especially concerning given the small number of participants and the potential for the effect to be significant in a larger sample.

3. An analysis of choice data or consistency (central to the argument of subjective value and task performance) is not reported as part of the paper.

4. Documentation of analysis needs additional help. A. Degrees of freedom are not reported. B. The model used and test performed are often unclear. Provide R model definitions directly in the paper to clarify each model and test performed. C. Post analysis scripts (custom, R, and FSL) on Github or a similar publicly accessible host.

5. What does the covariance table look like for the fMRI EVs? Is there a reason for models 1 or two independent of 3?

6. Arguing for neural differences from speed effects has the odd property of seeming to claim that the neural does not drive the behavioral. Which specific processes are claimed to be encoded in the brain? How do those processes produce the behavioral effects? This issue is especially clear during the analysis of different hypothesized reward signals in different spatial locations during which there is little explanation for which processes are believed to be separate and why.

Minor notes:

- Soften language like: "...our findings call for a reinterpretation of previous findings in research on value-based decision-making." or set in accordance with the path chosen for criticism 1.
- VD abbreviation not defined or used again later.

Reviewer #3:

Remarks to the Author:

There exists a large and ever-growing literature on value-based decision making. These findings incorporate concepts from many traditional fields, including psychology and neuroscience. This paper looks closely at two of those findings. One involves response times (RT) and one involves several regions of the brain frequently implicated in value-based decision-making. The authors propose an alternative explanation for these findings: goal congruency. They support their hypotheses with data from two studies, using both behavioral and fMRI data. The paper also includes some computational modeling results.

I am in general quite excited about the topic of this paper. It should be of broad interest to many academic communities and poses a valid challenge to some of the most canonical findings in the field. However, I have some reservations that would need to be addressed before publication.

The authors use a recent meta-analysis to motivate focusing on VMPFC and striatum. This is perfectly reasonable, but I wonder if the authors are leaving a lot on the table by not considering posterior cingulate cortex (PCC). I have two reasons for wondering this. One, the meta-analyses in the literature implicate PCC in value-based choice and also there is plenty of literature suggesting

the relevance of PCC for task-switching and adapting to one's environment. If the authors are going to claim analysis of the "valuation network" (Fig. 2) they will need to include PCC.

Several related notes about response times for the two tasks. I was somewhat surprised by the claim on page 147 that there were no RT differences. It is much more "natural" to choose the best item from a set than the worst. Wouldn't a main effect be expected? It would be helpful to see the average/median RTs for conditions/subjects, perhaps in an Appendix table. Also, from Fig. 1, is this to be interpreted as the task having sets where the overall value was zero? It would be helpful to also provide a table or figure with a distribution of what the overall set values were for the tasks in both studies.

Perhaps I have misunderstood some of the regressions, but how correlated are the regressors for overall reward value and overall goal value? Do we need to worry about multi-collinearity with those regressions and interpreting their results?

The authors definition of "value difference" is problematic. In 2AFC it is straightforward, but the authors have effectively mapped "range" onto "value difference" in a larger choice set. Wouldn't this assign the same "value difference" to set A of (5,1,1,1) and set B of (5,4,3,1)? This seems too limiting.

The paper could provide more details on what exact LCA model(s) were estimated, fit, and used for the simulations.

While I am not asking for any additional data, I have the following thought experiment for the authors. The story makes sense here, but what would happen if all aversive items were used? In theory, if the authors are correct, they should obtain similar results. Why not do this with aversive items? Would the authors expect it to work in that case?

Minor comments

- It would be better if the authors chose colors that are more decipherable when printing in greyscale (e.g. Fig 1 B-D).
- Line 212 says "regressions predicting BOLD activity" but this is not "prediction" is it? It is correlation.
- The work focusing on OFC as a "map" of "task space" (e.g. from Yael Niv and Geoff Schoenbaums' labs) seems relevant, doesn't it?
- Hawkins et al. (2014), "The best of times and the worst of times are interchangeable" might be of interest to the authors.

August 29, 2019

Dear Dr. Horder, dear reviewers,

We thank the reviewers for their thorough and helpful reviews. We have now substantially revised the manuscript in response to these comments, including extensive edits to the main text, additional analyses, computational model parameter estimation, and a follow-up experiment (Study S2) that addresses a concern raised by two of the reviewers (see Overall Response 3).

Below we address a few comments that were shared by more than one reviewer, by clarifying the study aims and terminology. We then go on to address each reviewer's individual comments in turn.

Sincerely,

Romy, Carolyn, and Amitai

Aims and terminology

Overall Response 1: Terminology

Reward vs. goal value. Throughout our paper we use *reward value* to refer to the expected reward or subjective value a person expects from acquiring an option (cf. Schultz, 2015). For instance, someone who places a higher reward value on beer than wine expects to enjoy the experience of having beer more than that of having wine. If they are given a menu with those two drink options and asked to choose the one they prefer more, they will choose beer. If they are asked to choose the one they prefer less, they will choose wine. In both cases, the reward value for beer always remains higher than the reward value for wine. *Goal value (or goal congruency)*, by contrast, refers to the extent to which an option's reward value supports one's current choice. In the situation above, beer has a higher goal value when being asked to choose the better item but wine has a higher goal value when asked to choose the less good item. The latter case is key to understanding the dissociation we are drawing - indicating that wine is less preferred doesn't make it any more desirable to consume (i.e., it is still associated with lower expected reward) but that choice is more congruent with the given task (i.e., choose worst).

Relative vs. overall reward/goal value. Options vary in the extent to which they are rewarding and the extent to which they are goal-congruent. If beer has a high expected reward then it is highly congruent with a choose-best goal and highly incongruent with a choose-worst goal. A set of options can therefore vary in how rewarding or congruent they are *relative to one another* (e.g., the extent to which beer is preferred to wine) and how rewarding or congruent they are *overall* (e.g., the extent to which an individual finds beer and wine to both be enjoyable). These examples also illustrate the fact that relative value and overall value are orthogonalizable (e.g., beer can be preferred to wine whether both have high value or low value, Figure S4).

The comments we received from Reviewers 1 and 2 clarified for us that we needed to not only explain how we were using these terms but to also distinguish our use of these terms from their other potential uses and from other closely related constructs:

Goal value is distinct from goal achievement. As Reviewer 1 suggests (comment 1), the terms goal value/congruency evoke the idea of goal *achievement*, which refers to the value one places on succeeding at their current goal (Gollwitzer & Sheeran, 2006). In this case, goal achievement can be equated with responding correctly on a given trial (i.e., choosing the best option in choose-best and the worst option in choose-worst). The likelihood of goal achievement should therefore scale with the *relative* goal value of an item (i.e., the degree to which that item's value is more goal-congruent than the others), which we likewise relate to the confidence one should feel when choosing that option. However, this is not the case for the *overall* goal value. That is, the likelihood of goal achievement (i.e., responding correctly) is not reliably affected by the items being all high goal value versus all low goal value (Study 1: $b = -.27$, $z = -0.90$, $p = .369$, Study 2: $b = 0.13$, $z = 0.48$, $p = .632$, Study S2: $b = -.44$, $z = -1.20$, $p = .230$). Our findings of behavioral and neural signatures of overall goal value are therefore not straightforwardly interpretable from a goal achievement perspective.

Goal value is distinct from context-specific subjective value. As Reviewer 2 suggests, the term goal value can also be interpreted as referring to the subjective value of an item within a particular context or motivational state. For instance, as the reviewer points out, the subjective value one places on a given food will reflect a multi-faceted utility function that weighs factors like motivational state (e.g., hunger level), features of the food (e.g., tastiness, healthiness), and context (e.g., the range of items on the menu). However, this multifaceted value function that has been well-characterized by Rangel, Glimcher, and others is what we are referring to as *reward* value, not goal value (cf. Introduction p. 4 l. 43 ff.). Across this literature, while subjective value is assumed to be sensitive to all of the factors described above, a basic assumption is that this value shouldn't reverse (i.e., it should remain directionally-consistent) if one is asked to indicate whether they like an item or whether they don't like it. In the beer/wine example, the reward values of each are no doubt multifaceted and may change on a given day or for a given meal but the reward value of the beer isn't expected to reverse if someone is handed a menu and asked to either (a) circle the option they want or (b) cross off the option they don't want. The kind of value that has this property of reversing with question framing is what we refer to as goal value/congruency.

Overall Response 2: Motivation and aims

The terminological clarifications above help to clarify the aims of our research. The decision neurosciences have typically assumed that reward values (and their multifaceted contributors) drive (a) response speed (approach or decision speed) and (b) neural activity in brain regions associated with reward processing (cf. e.g. Schultz, 2015). Our studies aimed to test these assumptions against an alternative hypothesis, grounded in cognitive science, that decision speed and neural activity are driven by the degree to which option values are congruent with an individual's choice goal.

Previous studies have been unable to probe this alternate hypothesis because the prevailing approach in neuroeconomics is to have participants indicate the option in a choice set that they most prefer (i.e., choose-best), the precise condition in which reward values and goal values are indistinguishable. We therefore sought to contrast these hypotheses by decorrelating these variables, having participants either choose the best option (where reward

and goal value are positively correlated) or choose the worst option (where these values are instead negatively correlated). In doing so, we were able to for the first time show that components of value-driven behavior and neural activity are only accounted for by goal congruency and not, as previously assumed, by reward value. These findings run counter to common assumptions in neuroeconomics and help to reshape existing theories regarding how rewards are processed and translated into choices.

Overall Response 3: Hypothetical vs. incentivized choices

Reviewer 1 and 2 raised the concern that our results may be an artifact of using hypothetical choices across the studies we report. To address this concern, we ran an additional study (Study S2, Supplemental Results 2), in which choices were incentivized in a manner analogous to standard neuroeconomic studies, with participants receiving one of the products in our set following the experiment. Paralleling previous studies, choosing an item in the choose best condition made it more likely to be selected as the bonus item; choosing an item in the choose worst condition made it *less* likely to be selected as a bonus item. The pattern of behavior we observed for these incentivized choices was identical to that observed in Studies 1 and 2. Critically, we again observe a significant interaction between task goal and Overall Reward Value ($b = -.41$, $p < .001$), such that the speeding effects of Overall Reward Value in the choose best condition ($b = -0.22$, $p = .001$) reverse in the choose worst condition ($b = .20$, $p = .006$), resulting in RT slowing instead. As in our other studies, we again found that our RT data was collectively better accounted for by Overall Goal Value ($b = -.20$, $p < .001$), with no residual effect of Overall Reward Value ($b = -.01$, $p = .837$). The consistency of these findings across hypothetical and non-hypothetical choice settings resonates with a broader literature that finds similar behavioral and neural patterns across the two settings (e.g., Kang, Rangel, Camus, & Camerer, 2011; compare also Shenhav & Buckner, 2014 [incentivized] with Shenhav, Dean Wolf & Karmarkar, 2018, Shenhav & Karmarkar, 2019 [hypothetical]), suggesting that participants are generally able to engage with the hypothetical choices as though they were making a choice that will be actualized.

Reviewer #1 (Remarks to the Author):

1. This is a well-designed and thoroughly performed and analyzed study. The methods are sound and state of the art and the inferences made by the authors are covered by the data. However, I am not sure how novel and surprising these results are and whether they signify an important advancement of our knowledge. In contrast to habitual behavior (which is not studied here), goal-directed behavior such as value-based or perceptual decision making is driven by the goals of the participant. I would not have expected that general value of a stimulus which does not have any relevance for the outcome of the task at hand but only serves as a feature dimension (similar as color, motion direction etc. in perceptual decision making) based on which a decision has to be made has a major influence on behavior. For participants it was irrelevant, whether they would like to have a stimulus or not, because the motivation and goals of participating in the task were completely uncoupled from the stimuli. It does not seem to matter, whether I have to make a judgment who of my colleagues wears the most or least expensive wearables or whether I discriminate the color of these wearables. "Reward value" might not be the best-suited term here: what the authors seem to mean is the value that participants would assign each item, had they the opportunity to get one of them. As

the authors state themselves, reward value is -at least motivationally- irrelevant for the participants in the task.

The actual reward participants expect from participating consists of many outcomes, including course credits/payment, doing this task at a minimal time, being perceived by the experimenter as a cooperative and intelligent subject etc. These rewards determine the (sub)goal of making the choices according to the instructions, which in turn drives decision making and behavior. The entire issue gets even clearer, when one imagines to transfer the task to a monkey lab. Monkeys would quickly learn that the values the associate with shown items (e.g., many grapes vs. one grape) are completely motivationally irrelevant, if their choice is rewarded only with a drop of water. Their behavior would not be influenced by pictures of rewarding items any more.

Thank you for your comment. We have now clarified our use of these constructs and how they fit into the broader literature. As we explain above, we use the term reward value to mean the value one assigns to acquiring an item, similar to how it is used in the reinforcement learning literature (e.g., Schultz, 2015) and similar to how the term 'acquisition utility' is used in the behavioral economics literature. We mean for this value to incorporate considerations of motivational state and context, as you suggest, but it is distinct from our use of the term goal value/congruency in an important way. Unlike reward value, the goal congruency of an item reverses when the task is to choose the worst rather than the best item.

The reviewer raises an important question, whether there is reason to think that *any* of the value signatures we observed reflect reward value (as we mean it) rather than only reflecting goal value. In other words, might our participants have selected between the items the way that an individual might make a perceptual or numerosity comparison, particularly given that choices were hypothetical in the studies we presented.

We have now addressed this question head-on by performing a follow-up study (Study S2, Supplemental Results 2) that was identical to Studies 1-2 except, unlike these previous studies, participant choices did influence whether they will receive a given product (identifying the item as the best vs. worst makes receiving it more vs. less likely, Supplemental Methods). This study successfully replicated the behavioral findings of our hypothetical studies.

As we note in our Overall Response 3, the similarity of our hypothetical and incentivized findings is consonant with a wider literature in neuroeconomics (e.g. Kang, Rangel, Camus & Camerer, 2011). This research has not only shown that reward-related behavior and neural activity are similar irrespective of hypotheticality but that the reward value of a stimulus is typically encoded by the reward circuit whether or not it is relevant to the current task (e.g. Kim et al, 2007; Lebreton et al., 2009). For instance, Tusche and colleagues (2010) scanned participants while they performed a cognitively demanding task and were told to ignore images of items that were shown in the background. Despite being task-irrelevant, the reward circuit tracked the value of those items and predicted preference-based choices between pairs of those items (all of which were assessed subsequent to the demanding task). In past work, we have similarly shown that this circuit tracks the overall value of a choice set (which is also motivationally-irrelevant to the task of choosing between items) both when the choice is incentivized (Shenhav & Buckner, 2014) and when it is hypothetical (Shenhav & Karmarkar, 2019).

Collectively, these findings suggest that people naturally encode the motivational relevance of stimuli even when hypothetical, and that this is facilitated by tasks like ours that consist of hundreds of unique items, minimizing habituation effects and making it very difficult for participants to choose based on abstract features of the stimuli (e.g., shape, color, quantity).

To address the concern about novelty, it is important to note that these past findings also make a straightforward prediction that higher activity in the reward circuit should predict a higher reward value for a given item, and such a coding scheme would be sufficient to perform our task. If a brain region encodes beer as having value 10 and wine as having value 1, there is no reason for it to reverse those values when choosing best vs. worst, rather than maintaining a directionally-consistent coding scheme and simply changing the operation to look for smallest rather than highest. From this perspective, our neural findings are particularly striking and in tension with past assumptions about this circuit. We therefore believe these goal value findings are both novel and of broad interest to the neuroscience community, particularly when juxtaposed with the simultaneous observation of neural correlates for reward value.

2. To me, the only surprising finding is that the overall reward value (i.e. the values they would assign to the items if they could get them, which, of course, is closely related to the set liking assessed in the post-trial survey) is also represented in the dorsomedial striatum. Of course, to do the task, the values of the items have to be retrieved, such that their representation in vmPFC could be expected, but the fact that the striatum, a structure related to learning and decision making, also represents them may be less expected. Could it be possible that the representation of the overall reward value was actually related to the anticipation of the upcoming post-choice affective rating of set appraisal?

We agree that relevance for an upcoming task could lead participants to process information related to that task while performing the current one. However, in our study, participants were, by design, unaware that they would be subsequently asked to rate the overall Liking of the sets. Our findings further mirror previous work on automatic reward representations (e.g. Kim et al, 2007; Lebreton et al., 2009), as well as those of a recent study from our lab testing exactly this (Shenhav and Karmarkar, 2019).

3. Related to this, the exact timing of stimulus presentation and choices should be reported in more detail (at least in the supplemental information).

Thank you for pointing this out. We previously reported the inter-trial-interval for the scanning (2-7s, uniformly distributed), and that there were no response deadlines in any of the phases of the experiment (Supplemental Methods). We now added that the choice stimuli were displayed on the screen until a choice was made, for completeness.

4. Generally, the authors may be correct that in previous studies many researchers have overlooked the potential difference between values of an item and the actual outcome of an action (i.e. goal achievement). So, perhaps it might be a good idea to clarify this point which every good experimenter should always keep in mind. It is generally important to be as clear as possible with the terms used in the manuscript. Can one really talk about "reward value" and "preferred" option in the present task in the strict sense?

We thank the reviewer for this suggestion. We have now expanded on the implications of our findings for distinguishing between constructs being manipulated and measured in decision-

making research (pp. 23). We have also clarified our terminology throughout the manuscript, including being more explicit about the distinction between the variable we refer to as goal value and the idea of goal achievement, where the former scales with the value of each individual item and the latter scales with the likelihood of responding correctly (and is therefore only connected to the *relative* value between items). As we enumerate in Overall Response 1, the effects of overall goal value on behavioral and neural data therefore cannot be accounted for by goal achievement.

We believe that our substantially revised explanation of what we are referring to as goal congruency/value will avoid readers drawing an association between this and goal achievement (cf. p.6).

5. The striatal gradient reported for the overall reward value is interesting. I wonder whether and -if so- how this fits to a recent study in which a striatal gradient in model updating based on simple reinforcement learning vs. long-term optimal inference was found (Fischer, Bourgeois-Gironde, Ullsperger, Nat Commun, 2017). There, participants experienced incongruency of (long-term) goal and short-term reward. Optimal value updates guiding their upcoming choices in the service of the long-term goal were represented most strongly in mediodorsal striatum (caudate) whereas the (short-term) reward prediction error biasing their choice towards suboptimal behavior was most strongly represented in the ventral striatum. Can this be reconciled with the present results?

Thank you for pointing us to this paper. We do indeed think our results are consistent with those and now included this in our discussion. In their study, like in ours, information relevant to the current trial (short-term reward) was represented more ventrally, whereas information relevant to long term reward was represented more dorsally. Intriguingly, consistent with our interpretation that dorsal activation associated with overall reward value may reflect increased control demands (p. 23), the study further found that the encoding of short-term reward in the more dorsal regions was associated with the extent to which this information detracted from the participant's goal of inferring long term rewards.

6. Please provide more detail about the ROI. A figure showing the extent of the vmPFC and striatal regions would be quite informative to the readers, particularly those who are less familiar with the field.

Thank you for the suggestion. We now include maps of the Bartra ROI in Figure 2.

7. I would also find it interesting to see the same subregional analyses provided for the striatum done for the vmPFC.

Following up on your suggestion, we ran an analysis similar to that reported in Shenhav and Karmarker (2019) to test for differential effects in subregions of vmPFC (rostral ACC vs. medial OFC, Supplemental Results 6). Unlike in the striatum, we did not find an interaction between subregion and value effects and therefore avoid concluding that these regions code different information. However, for completeness we have now reported the trends we observe across these vmPFC subregions -- with higher overall reward value signals in rACC and higher overall and relative goal value signals in mOFC -- in the supplement.

8. Please provide more details on the fitting of the LCA models. What exactly is meant by "manual fitting"?

We apologize for not clarifying this earlier. Because our goal is primarily to demonstrate the divergent predictions each model makes, we previously tested a set of plausible parameters before settling on ones that were qualitatively in line with our empirical findings (RT and accuracy) but we had not performed quantitative fits to identify a global minimum in our cost function. Based on the reviewer's suggestion, we have now performed such quantitative fits by performing gradient descent on this same cost function. We report the details of this fitting procedure in the Supplementary Methods (SI, p.6). When simulating expected findings using the newly identified best-fit parameters, we arrived at the same pattern as previously reported (FIG. 1D, p. 13). We additionally verified that this pattern of predicted findings was robust to variations in parameter values.

Very minor:

9. p. 9, abbreviation VD was not introduced beforehand

Thank you for pointing this out. We corrected this mistake.

10. SI, p. 2 l. 28: doubled "were recruited"

Thank you. We corrected this typo.

Reviewer #2 (Remarks to the Author):

1. The overall tone of the paper is one of redefining what reward is and whether it is task-defined or more generally part of an approach-avoid system (described as Pavlovian in the text). The paper makes mixed use of different descriptions of utility, context, and traditional approach system distinctions (Pavlovian vs. Operant) while not fully addressing the literature in any of these domains, making it very challenging to understand the implications of the work or what exactly should be reinterpreted. The best option moving forward would seem to be to focus on context effects on value (see work by Rangel; Lee and Sul, Neuron; Clithero et al Choice independent value etc.). The manipulation from this paper could be seen as another contributor to a multifaceted utility function through which context, local goals, and other factors contribute to option selection.

We actually did not aim to redefine what reward is, but we can see how our previous Introduction section may have given that impression. As we now clarify in our revised manuscript and elaborate on in Overall Response 1 above, we in fact use *reward* value to refer to the context-sensitive, state-sensitive, multifaceted utility of an item (p. 4). When a participant saw a given item in our experiment, we assume that the reward value they assigned to it at the start of the experiment reflected a variety of factors that have indeed been elaborated on in previous research by Rangel and others.

Critically, while previous studies demonstrate factors that determine how much reward is placed on an item, a basic assumption underlying this literature is that this reward value shouldn't change dramatically (and certainly shouldn't reverse) if a person is asked to indicate

whether they like something the least or the most out of a set. As we discuss above, indicating that you like wine less than beer shouldn't make wine more rewarding for you than beer. Not only should reward value intuitively stay the same whether asked whether something is more preferred or less preferred, for it to reverse across these conditions would be maladaptive for the animal. This sets reward value (multifaceted and context-sensitive though it might be) apart from what we refer to as *goal value* or *goal congruency*.

Goal value refers to the extent to which reward value supports your current task goal, and therefore will reverse flexibly depending on whether higher or lower reward values align with the choice you need to make on a given trial (best vs. worst). The goal/task-sensitivity we are studying is therefore distinct in several important ways from the finds of context/state-sensitivity that have previously been studied in the decision-making literature, in that we are not studying how features of the environment alter reward value but rather how the environment (task) alters the way in which rewards are processed.

2. Relative weights of task and general rewards. The paper works to emphasize the role of task-goals but there are several potential issues:

A. Hypothetical choices reduce the magnitude of representation expected from the choice options independent of goal.

We have now addressed this concern with an additional behavioral study in which choices are not hypothetical (Study S2). Replicating our previous studies, behavior is only accounted for by goal value and not reward value, suggesting that the hypotheticality of those earlier studies did not qualitatively alter the relative processing of reward vs. goal value. Please also see the general response above.

B. RT model comparison analysis shows borderline marginal effect of reward.

It is disingenuous to describe this as showing effects exclusive to task even if the effects is not (strictly speaking) significant. This is especially concerning given the small number of participants and the potential for the effect to be significant in a larger sample.

Thank you for pointing this out. It is true that in addition to the value effects of interest (overall goal value, $b = -0.37$, $p < 2e-16$,) we found non-significant residual effects of overall reward value in Study 1 ($b = 0.05$, $p = .107$). However, the interpretation of this residual effect is complicated by two additional considerations. First, this trend for an overall reward value is the *opposite* of what has been reported in all previous work (i.e., slowing rather than speeding with increasing overall value) and therefore the opposite of what would be predicted by the reward value specific theories we highlight in the introduction. Second, this overall reward value effect did not replicate in Study 2 ($b = -0.01$, $p = .809$) or in the new incentivized study we have added (Study S2; $b = -0.01$, $p = 0.837$) (p. 11,13).

3. An analysis of choice data or consistency (central to the argument of subjective value and task performance) is not reported as part of the paper.

We apologize for having been unclear about the analyses we ran. We did previously report the equivalent of choice consistency findings, referring to these as reflecting choice "accuracy":

Across both studies, we find that value difference also predicts the accuracy of a given choice – as value difference increased, participants were more likely to choose the item

that achieved their current choice goal (the highest-value item in Choose Best, the lowest value item in Choose Worst; $b = 0.37$, $z > 13.14$, $p < .001$). The influence of value difference on choice accuracy did not differ between the two choice goals ($b < 0.58$, $z < 1.01$, $p > .311$). Unlike value difference, the overall value of a set did not predict choice accuracy for either task ($b < 0.49$, $z < 1.70$, $p > .089$). (p. 10 of original ms)

We have now replaced “accuracy” with “consistency” for clarity, and also report the statistics separately for study 1 and 2 (paralleling the edits described above for our RT analyses).

4. Documentation of analysis needs additional help. A. Degrees of freedom are not reported. B. The model used and test performed are often unclear. Provide R model definitions directly in the paper to clarify each model and test performed. C. Post analysis scripts (custom, R, and FSL) on Github or a similar publicly accessible host.

Following your suggestion, we have uploaded our analyses scripts to a github repository for public access <https://github.com/froemero/goal-congruency-dominates-reward>. This code/document has all model definitions and outputs including Kenward-Rogers approximated degrees of freedom. Degrees of freedom are now further added to the tables, most of which we moved to the main text. In the main text we now further clarify the statistical tests we ran.

5. What does the covariance table look like for the fMRI EVs?

By design, all of our regressors are uncorrelated. To visualize this, we added a figure with value distributions for all value regressors and a correlation table in the Supplement (Figure S4). The table shows that our manipulation was successful with regard to the independence of our regressors.

Is there a reason for models 1 or two independent of 3?

Models 1 and 2 serve to directly test our competing hypotheses about which values drive brain activity in the valuation network. We now clarify our motivation for model 1 (with only reward values) and model 2 (with only goal values) before reporting the results (p. 15).

6. Arguing for neural differences from speed effects has the odd property of seeming to claim that the neural does not drive the behavioral. Which specific processes are claimed to be encoded in the brain? How do those processes produce the behavioral effects? This issue is especially clear during the analysis of different hypothesized reward signals in different spatial locations during which there is little explanation for which processes are believed to be separate and why.

These are important questions and we apologize for not addressing them more directly in our original submission. In our revised manuscript, we have clarified and expanded on our previous explanation of the processes we believe may underpin our neural findings, and how they relate to previous interpretations of this network’s involvement in decision-making (pp. 20-22). Briefly, a large body of past work suggests that the reward circuit accumulates reward values (as defined above) in order to arrive at a choice. Our study distinguishes this process (evidence accumulation) and representation (reward value), and suggests that these may engage the circuit for different reasons.

The goal congruency signals we observe (relative and overall goal value) can be interpreted in one of two ways:

1) The most straightforward interpretation is that these signals reflect the accumulation of goal-relevant information. This is consistent with the process previously assumed to occur in this circuit (e.g. Rangel & Hare, 2010; Kable & Glimcher, 2009) but *inconsistent* with the information content that this circuit was assumed to be accumulating (i.e., reward value). Under such an account, relative and overall goal value collectively determine the total information accumulated for a response; neural activity provides a read-out of this accumulation process; and this process culminates in the response (cf. Hunt et al., 2012). This would explain why goal value variables predict choices and RT.

2) It is also possible for neural signals of relative goal value (the stronger of the goal value signals we observe) to reflect metacognitive monitoring of the decision process (i.e., as a read-out of *confidence* in one's decision) rather than (or in addition to) reflecting the decision process itself (Lebreton et al, 2015). Under this account, RT and neural activity would both have a common cause (evidence strength) but neural activity would not be determining RT.

Because it does not predict behavior (and therefore is not associated with evidence accumulation), the additional neural signal we find to be associated with overall reward value cannot be explained by either of the accounts above. However, it too has two plausible interpretations:

1) The most straightforward interpretation is that this signal simply reflects an appraisal of how rewarding the current environment is but does not directly influence the decisions being made in the scanner. This interpretation is consistent with our previous studies demonstrating (in a choose-best context) that this circuit signals such appraisals relatively automatically and independently of choice-related signals (Shenhav & Karmarkar, 2019, Sci Rep; Froemer & Shenhav, 2019, BioRxiv).

2) Because reward value is task-irrelevant but is motivationally-relevant, it is possible that some regions track these signals as an indication of the cognitive demands of needing to suppress these reward signals. Consistent with a study cited by Reviewer 1 (Fischer, Bourgeois-Gironde, & Ullsperger, 2017), we propose that this account may be particularly relevant for interpreting the ventral-dorsal distinction we observe in striatum (pp. 23).

Minor notes:

- Soften language like: "...our findings call for a reinterpretation of previous findings in research on value-based decision-making." or set in accordance with the path chosen for criticism 1.

We now revised our language throughout the manuscript to address the comments above. We elaborate on this specific example in our response above to this reviewer's comment.

- VD abbreviation not defined or used again later.

Thank you, we corrected this oversight.

Reviewer #3 (Remarks to the Author):

1. The authors use a recent meta-analysis to motivate focusing on VMPFC and striatum. This is perfectly reasonable, but I wonder if the authors are leaving a lot on the table by not

considering posterior cingulate cortex (PCC). I have two reasons for wondering this. One, the meta-analyses in the literature implicate PCC in value-based choice and also there is plenty of literature suggesting the relevance of PCC for task-switching and adapting to one's environment. If the authors are going to claim analysis of the "valuation network" (Fig. 2) they will need to include PCC.

Thank you for your suggestion. Following your suggestion we analyzed PCC and found that it tracks Overall Reward and Relative Goal Value, paralleling our findings in vmPFC and striatum. We now report these results in the supplement (Supplemental Results 4, Table S4).

2. Several related notes about response times for the two tasks. I was somewhat surprised by the claim on page 147 that there were no RT differences. It is much more "natural" to choose the best item from a set than the worst. Wouldn't a main effect be expected?

We can understand why choosing the worst item may seem more natural than choosing the best item, and would not have been surprised if that turned out to be the case for our study. However, only one of our studies (Study 1) observed such as (non-significant trend). We speculate that we did not observe this overall slowing effect because the tasks were performed in blocks (choose-best in one block and choose-worst in the other). The unnaturalness of the task may therefore have been mitigated through repetition over the same task set.

It would be helpful to see the average/median RTs for conditions/subjects, perhaps in an Appendix table. Also, from Fig. 1, is this to be interpreted as the task having sets where the overall value was zero? It would be helpful to also provide a table or figure with a distribution of what the overall set values were for the tasks in both studies.

We uploaded scripts and a compiled analyses output on a github repository in which we show by-condition RT summary and distribution plots. We now further added value distribution plots to the Supplement (Figure S4). We indeed used the entire range of values consistent with Shenhav et al. (2018).

3. Perhaps I have misunderstood some of the regressions, but how correlated are the regressors for overall reward value and overall goal value? Do we need to worry about multicollinearity with those regressions and interpreting their results?

Thank you for your comment. The studies were designed such that these regressors are orthogonal, which we now clarify in the revised manuscript (p. 15). As we now also show in value distribution plots and a regressor covariance table in the Supplement (Figure S4), across both tasks, the regressors are completely uncorrelated.

4. The authors definition of "value difference" is problematic. In 2AFC it is straightforward, but the authors have effectively mapped "range" onto "value difference" in a larger choice set. Wouldn't this assign the same "value difference" to set A of (5,1,1,1) and set B of (5,4,3,1)? This seems too limiting.

We apologize for not describing these calculations more clearly. We calculate value difference as the absolute difference between the goal (max for choose best and min for choose worst) and the average of the remaining values. Using your example, the value difference for the first set would be $5 - \text{mean}(1,1,1) = 4$. The value difference for the second set would be 5-

$\text{mean}(4,3,1) = 2.333$. Thus the value difference in the first example would be higher than the value difference in the second example. In this and previous studies (Shenhav & Karmarkar, 2019; Shenhav, Dean Wolf & Karmarkar, 2018) we compared this measure of value difference to plausible alternatives (e.g. max goal value minus the goal value of the next best item) and found that our chosen measure provided the best account of our RT data. We revised this line to make it more clear.

5. The paper could provide more details on what exact LCA model(s) were estimated, fit, and used for the simulations.

We apologize for not clarifying these details earlier. In the Supplementary Methods (p. 6), we now expand further on the models, model simulations, and the new model fitting procedure added to estimate parameters for the simulations in the revised manuscript (see Fig. 1D, p. 14)

6. While I am not asking for any additional data, I have the following thought experiment for the authors. The story makes sense here, but what would happen if all aversive items were used? In theory, if the authors are correct, they should obtain similar results. Why not do this with aversive items? Would the authors expect it to work in that case?

Based on previous research on decision-making in the domains of positive and negative outcomes, we would expect to find similar neural signals of goal value when choosing between aversive options as we found when choosing between appetitive items. However, a potential difference is that we may find additional regions tracking overall (negative) value, for instance the anterior insula. We think both of these would be interesting predictions to test, and have now proposed this as an important direction for future research (p. 24).

Minor comments

- It would be better if the authors chose colors that are more decipherable when printing in greyscale (e.g. Fig 1 B-D).

Thank you for your suggestion. We have now changed the colors to be easier to distinguish in greyscale.

- Line 212 says “regressions predicting BOLD activity” but this is not “prediction” is it? It is correlation.

Thank you, we changed this wording.

- The work focusing on OFC as a “map” of “task space” (e.g. from Yael Niv and Geoff Schoenbaums’ labs) seems relevant, doesn’t it?

We agree that this is a valuable connection. We now mention a potential link between our findings and previous work tying mOFC to representations of task space representations (p. 21).

- Hawkins et al. (2014), “The best of times and the worst of times are interchangeable” might be of interest to the authors.

Thank you for this excellent recommendation! We now cite this publication in the discussion (p. 21).

Reviewers' Comments:

Reviewer #1:

Remarks to the Author:

Thanks to the authors for the clear reply and the thorough revision. The clarifications of their nomenclature and research goals and the additional experimental data have changed my opinion significantly. I now see the novelty of their findings and the importance of the manuscript for the field (which the authors also clarified in their revised version).

I have no further comments or suggestions. In my view the manuscript is ready for publication.

Reviewer #2:

Remarks to the Author:

The authors have gone through a great deal of effort to improve their manuscript. Congratulations! While it is not perfect, it does make a significant contribution to the literature. The last item that should be addressed is the public availability of the data and script set. Neither is currently accessible (the GitHub link in the manuscript gave a 404). The other data should be added to openneuro or something similar.

Reviewer #3:

Remarks to the Author:

Authors have done an excellent job addressing the concerns put forth by the three reviewers.

I have two remaining questions/issues.

It was interesting to see the authors reference "acquisition utility" in the response to R1. I would ask them to think through this carefully and if they do feel this is in line with that literature in behavioral economics, they should add a reference in the main paper.

Thank you for looking into the PCC. There are some interesting results and I am glad the authors include them. Looking at Tables S1 and S2, though, I am curious since PCC seems to have equally if not maybe larger effect sizes. Can the authors run some comparisons to see how the three regions (VMPFC, VSTR, PCC) shake out? It would be very interesting to see how they compare in a more formal setting. Also, given the results why not include PCC in Figure 2?

September 13, 2019

Dear reviewers,

We are very happy that the reviewers found our manuscript much improved. We address the remaining questions from Reviewer 3 below.

Thank you again for your time and attention.

Sincerely,

Romy, Carolyn, and Amitai

Reviewer #3:

It was interesting to see the authors reference "acquisition utility" in the response to R1. I would ask them to think through this carefully and if they do feel this is in line with that literature in behavioral economics, they should add a reference in the main paper.

Thank you for this suggestion. After careful scrutiny of the relevant literature, we have concluded that it is best to refrain from drawing an explicit connection between *reward value* (as we are using it) and *acquisition utility*. While the latter term has been used to loosely refer to the value of options, there is a more precise definition in the literature that also takes into account the cost of acquiring an option (Thaler, 1999, Journal of Behavioral Decision Making). Given that our data cannot speak to this cost component, we would prefer to remain agnostic on this front and maintain a more inclusive definition of reward value. We have therefore chosen to continue to omit the citation from the manuscript in order to avoid confusion and/or misinterpretation. We apologize for not being more precise in our use of this terminology in the previous response letter.

Thank you for looking into the PCC. There are some interesting results and I am glad the authors include them. Looking at Tables S1 and S2, though, I am curious since PCC seems to have equally if not maybe larger effect sizes. Can the authors run some comparisons to see how the three regions (VMPFC, VSTR, PCC) shake out? It would be very interesting to see how they compare in a more formal setting. Also, given the results why not include PCC in Figure 2?

This is a great question! We ran an additional analysis to whether there were any significant differences between how these three regions encode our variables of interest, but did not find any (interactions between region and value variable p s > 0.20; see table below). We also appreciate the suggestion of incorporating information about PCC into Figure 2. We are refraining from incorporating the PCC results because, unlike our analyses of the Bartra network, the PCC analysis was post-hoc. However, we do think the readers would find it helpful if we visualized the location of our PCC mask as we have for the Bartra network in Fig. 2A. We have therefore added a figure (Figure S6) that juxtaposes our PCC mask and the two regions that comprise the Bartra network (vmPFC, vStr), which were also separately analyzed in post-hoc analyses.

	Sum Sq	Mean Sq	NumDF	DenDF	F value	Pr(>F)
Best-Worst Condition	0.16	0.16	1	34.48	0.06	0.81
OV _{reward}	52.33	52.33	1	11607.1	19.19	0
RV _{reward}	2.1	2.1	1	12582.98	0.77	0.38
OV _{goal}	15.18	15.18	1	5199.24	5.57	0.018
RV _{goal}	79.35	79.35	1	12315.49	29.1	0
Region	76.16	38.08	2	29.69	13.96	0
RT	1.87	1.87	1	30.73	0.69	0.414
C _{b,w} :Region	8.07	4.04	2	12665.97	1.48	0.228
OV _{reward} :Region	0.79	0.4	2	3859.67	0.15	0.864
RV _{reward} :Region	2.8	1.4	2	12684.96	0.51	0.598
OV _{goal} :Region	6.32	3.16	2	12673.22	1.16	0.314
RV _{goal} :Region	5.61	2.8	2	12527.73	1.03	0.358

Reviewers' Comments:

Reviewer #3:

Remarks to the Author:

Congrats to the authors on a fine paper. Looking forward to seeing it in print.